# Design and Implementation of an Inductive Proximity Sensor with Embedded Systems

**DOI:** 10.3390/s25196258

**Published:** 2025-10-09

**Authors:** Septimiu Sever Pop, Alexandru-Florin Flutur, Alexandra Fodor

**Affiliations:** Applied Electronics Department, Faculty of Electronics, Telecommunications and Information Technology, Technical University of Cluj-Napoca, 400027 Cluj-Napoca, Romania; septimiu.pop@ael.utcluj.ro (S.S.P.); alexandru.flutur@ael.utcluj.ro (A.-F.F.)

**Keywords:** inductive proximity sensor, displacement measurement, LC oscillator, frequency, embedded

## Abstract

Non-mechanical contact distance measurement solutions are becoming more and more necessary in various industries, including building monitoring, automotive, and aviation industries. Inductive proximity sensor (IPS) technology is becoming a more popular solution in the field of short distances. Because of its small size, dependability, and measurement capabilities, IPS is a good option. Separate circuits are used in the classical structures to generate the excitation signal for the sensor coil and measure the response signal. The response signal’s amplitude is typically measured. This article proposes an IPS model that uses frequency response as its basis for operation. A microcontroller and embedded technology are used to implement a small IPS structure. This includes the circuit for determining distance, as well as the signal generator used to excite the sensor coil. In essence, an LC circuit is employed, which at the unit step has a damped oscillatory response by nature. Periodically injecting energy into the LC circuit, however, causes it to enter a persistent oscillatory state. The full experimental model is implemented and presented in the article, illustrating how the distance can be measured with a 33 µm accuracy within the 10 mm range with the help of the nonlinear relationship between frequency and distance and the linear drift of frequency with temperature.

## 1. Introduction

Oscillators are essential circuits in modern electronics, supporting everything from precision measurement and signal processing to communication systems with steady periodic signals. The incorporation of oscillators into sensing has gathered considerable attention due to their natural simplicity, high sensitivity, and capability to function in real time. These oscillators function by responding to changes in their resonant frequency, which can be directly affected by intrinsic circuit elements or characteristics of the surrounding materials, such as permittivity, permeability, or conductivity [1,2,3]. By integrating or exposing reactive components—usually inductors or capacitors—to the material being analyzed, variations in the oscillation frequency can be linked to distinct physical, chemical, or structural attributes. This sensing mechanism facilitates a broad range of applications, from the identification of materials to bio-sensing, chemical detection, and the monitoring of structural conditions [4,5,6,7].

Throughout the literature, several works employ oscillator sensing systems for various applications. In [8], Zhao et al. propose an inductive proximity sensor having an attenuation coefficient of 1. It combines differential inductance detection with querying proximity tables that are stored in a field-programmable gate array (FPGA) to detect target metal objects located at the same inductive distance, proving that the approach is accurate and effective.

The inductive sensing principle is utilized by Ripka et al. in [9] to develop eddy current inductive speed and position sensors that can work up to 20 m, the authors achieving a 0.4% linearity for the linearity error of ferromagnetic targets. Sensors based on eddy currents are used also in controlling the quality of materials, as the non-destructive technique presented in [10] shows. Capacitive sensing is employed in [11] for the implementation of a smart sensor for monitoring respiratory rate. Furthermore, the capacitive sensor is incorporated into a Colpitts oscillator, achieving a high-quality factor and frequency stability. Also, the authors prove that the parasitic capacitances that appear in the circuitry itself can be minimized. The capacitive sensing method is utilized also in [12] to develop a mass sensor, where an LC circuit is used in conjunction with a frequency demodulator and frequency counter. Using the proposed capacitive detection, the detectable minimum value for a mass of below 1 × 10^−14^ g was obtained with a time constant of 0.08 s. Evidently, more popular applications for inductive sensing such as temperature sensing, as well as speed and position sensing, can be employed using inductive elements [13,14].

The usage of a sensor-to-microcontroller interface is far from new. In [15], a sensor-to-microcontroller interface circuit is presented, one that lacks the need for a calibration inductor. The suggested approach is predicated on an RL circuit’s time response. It is possible to estimate the sensor inductance by using the time constant of the voltage across the inductor. For this reason, this method is particularly applicable to sensors with a variable self-inductance.

The measurement method for the eddy current-based inductive sensors employs a sinewave [16]. A magnetic field is produced when the sensor’s coil is stimulated by a sinusoidal signal. A metallic object placed in a magnetic field will produce an induced current known as an eddy current, according to Faraday’s Law of Induction. A non-electric physical quantity, like temperature or distance, is transformed into inductance by the inductive sensor using the eddy current effect. We refer to those sensors as inductive proximity sensors, or IPSs. Due to the eddy current effect, the balanced or unbalanced bridge topology is the most used conditioning circuitry [17]. Colpitts or Harley oscillators are examples of sinusoidal oscillators that can also be used. Numerous electronic components are used in intricate structures that make up both types of conditioning circuitry [18,19].

In line with the previous studies reported in the literature [20,21,22], the proposed IPS can be applied to precision displacement measurement tasks, benefiting from its high resolution and wide measurement range. Compared to all the previously mentioned research, our research aims to present an efficient measurement technique for capacitive or inductive sensors, using an embedded system that uses the changes in the oscillator’s resonant frequency for sensing. Due to its versatility, it can be adapted to both capacitive and inductive sensing, but, given the fact that an inductive sensor is more complex, the use case presented below is for the latter.

The paper is structured as follows. Section 2 presents the materials and methods used to demonstrate the utility of the current configuration, along with the mathematical model for determining the value for the inductance sensor. The study started with a parallel LC circuit time step response combined with the conditioning circuit employing a microcontroller. Section 3 presents the results and measurements performed with the proposed system, followed by Section 4, in which several conclusions are drawn and discussed.

## 2. Materials and Methods

### 2.1. IPS Conditioning Circuitry Analysis Suitable for Embedded Technology

To analyze the circuit, the inductive proximity sensor is modelled with its inductivity, L_o_, and its equivalent series resistance, r_o_. The two circuit parameters depend on the coupling (therefore, the distance) and temperature [13]. In parallel with the coil, the capacitor C_P_ is connected, and the circuit L_o_–r_o_–C_P_ forms a resonant circuit (tank circuit). When a metallic object is placed in the vicinity of the coil, part of the electromagnetic field energy produced by L_o_ is transferred to the conductive object. Due to energy transfer, a decrease in inductance and an increase in resistance is obtained. The electrical circuit proposed for distance sensing is shown in Figure 1.

The capacitor C_P_ has a fixed value, chosen to be much higher than the parasitic capacity of the coil. Considering that C_P_ and L_o_ are in parallel, the parasitic capacity of the inductor can be neglected. The driving logic with the processing unit forms the loading circuit for the sensing coil. The resistance R, together with the coil’s own resistance r_o_, limits the charging current of the coil L_o_, responsible for creating the magnetic field.

In Figure 2, the equivalent sensor circuit is presented, having the command signal *u*_D_(t), depending on the state of the Q transistor in Figure 1. From the perspective of the circuit formed by the L_o_, r_o_, and C_p_, the applied signal *u*_D_(t) is a step input voltage. The first loop contains C_P_, R, and *u*_D_(t) and the secondary loop L_o_, r_o_, and C_p_. If we apply Kirchhoff’s voltage law, the equation system (1) is obtained.(1)$$\left\{\begin{array}{c} R\left(i_{C}+i_{L}\right)+u_{C}=u_{D,}\\ r_{o}i_{L}+u_{L}-u_{C}=0. \end{array}\right.$$

Replacing the voltage–current relations into Equation (1), the differential equations of the equivalent circuit are as follows:{(2a)RCPduCdt+iL+uC=uD(2b)roiL+LoduLdt−uC=0

The term *i_L_* in the above equation should be eliminated by substituting it from Equation (2a) into Equation (2b). This substitution yields a second-order differential equation, with the output voltage at the measurement point *u_C_* as the variable.(3)$$RL_{o}C_{P}\frac{d^{2}u_{C}}{{dt}^{2}}+\left(Rr_{o}C_{P}+L_{o}\right)\frac{du_{C}}{dt}+\left(R+r_{o}\right)u_{C}={r_{o}\;u}_{D}$$

The general solution of the nonhomogeneous equation in (3) is the sum of the general solution ${u_{C}\;\left(t\right)}_{H}\;$ of the related homogeneous equation and the particular solution ${u_{C}\;\left(t\right)}_{P}\;$ of the nonhomogeneous equation, as expanded in (4):(4)$$u_{C}\;\left(t\right)=\;{u_{C}\;\left(t\right)}_{H}\;+{u_{C}\;\left(t\right)}_{P}$$

The roots of the characteristic equation are as follows:(5)$$\lambda_{1\;\;1,2}=\frac{-\left(Rr_{o}C_{P}+L_{o}\right)\pm \sqrt{{\left(Rr_{o}C_{P}+L_{o}\right)}^{2}-4RL_{o}C_{P}\left(R+r_{o}\right)}}{2RL_{o}C_{P}}$$

The waveform response of the output signal at the measurement point $u_{C}\;\left(t\right)$ depends on the solutions of characteristic Equation (5). There are three cases to consider: when the two roots are distinct and real, the output signal response will be overdamped. Initially, the voltage at the measurement point will increase due to the capacitor charging and then decrease due to the inductor charging. If the roots are real and equal, the second-order system response will be critically damped. Finally, if the roots are distinct, complex conjugates, the second-order system response will be underdamped, and the output signal will exhibit damped oscillations.

All parameters of the electronic components are real, positive values. To achieve an underdamped waveform for the output signal, we will analyze the influence of the sensing coil inductance L_o_ and the capacitance C_P_ on the system’s behavior. The circuit must satisfy the following conditions.
(6)$${\left(Rr_{o}C_{P}+L_{o}\right)}^{2}-4RL_{o}C_{P}\left(R+r_{o}\right)<0,\;\mathrm{or}\;{f\left(L_{o}{,C}_{P}\right)=\left(Rr_{o}C_{P}+L_{o}\right)}^{2}-4RL_{o}C_{P}\left(R+r_{o}\right)<0$$

The inductance of the sensing coil is not a constant parameter; in fact, its value changes as the detected material moves closer. Equation (6) is an upward-opening parabola that defines the maximum variation in inductance required to achieve underdamped behavior, as illustrated in Figure 3. Both ends of the inductance range are determined by the zeros of Equation (6), with these points corresponding to the solutions of Equation (6).(7)$$\left\{\begin{array}{c} L_{o,1}=C_{P}R\left(2R+r_{o}\right)-2\sqrt{R\left(R-r_{o}\right)}\\ L_{o,2}=C_{P}R\left(2R+r_{o}\right)+2\sqrt{R\left(R-r_{o}\right)} \end{array}\right.$$

The inductance L_o_ should be found in the interval [L_o,1_, L_o,2_], both extreme points being dependent on the capacitor C_P_ value; a low C_P_ value would reduce the graph area where the second-order system would have an underdamped response. Outside the region marked in Figure 3, Equation (6) is positive, $∆\left(L_{o}{,C}_{P}\right)>0$, and the second-order system will exhibit overdamped behavior.

Furthermore, the values of the electronic components were selected to ensure compliance with the conditions expressed in Equation (6). The core element of the equivalent circuit is the inductance of the sensing coil. In our experimental setup, the initial inductance of the sensing coil is 4.7 mH, exhibiting a temperature drift of 350 ppm/°C. The inductance value decreases by up to 10% from its initial value, L_o_. The equivalent series resistance of the coil, r_o_, is 16 Ω, with a temperature drift of 3900 ppm/°C. The coil used in our experiment has a cylindric geometry with 5.5 mm internal diameter and 8.5 mm external diameter; it is made with a copper wire of 0.16 mm diameter making 350 turns, on a ferrite core. The parallel capacitor C_P_ has a value of 22 nF, with a tolerance of 5%. This capacitance is significantly larger than the measured parasitic capacitance of the sensing coil, approximately 15 pF, allowing the parasitic effects to be neglected. The circuit’s current-limiting resistor R was selected as 1 kΩ to ensure that the current remains within the maximum allowable limit specified by the manufacturer.

Considering the damping effect introduced by the resistance in the circuit, the solution to the system of equations can be determined based on the damped response:(8)$$\left\{\begin{array}{c} {u_{C}\;\left(t\right)}_{H}\;=e^{\propto_{1}t}\left(A_{1}\cos\left(\omega_{d1}t\right)+B_{1}\sin\left(\omega_{d1}t\right)\right)\\ where\;\propto_{1}=-\frac{1}{2}\left(\frac{r_{o}}{L_{o}}+\frac{1}{RC_{P}}\right),\;\omega_{d1}=\sqrt{\frac{R+r_{o}}{{RL}_{o}C_{P}}-{\left[\frac{1}{2}\left(\frac{r_{o}}{L_{o}}+\frac{1}{RC_{P}}\right)\right]}^{2}}\\ or\;\omega_{d1}=\sqrt{{\omega_{1}}^{2}-{\propto_{1}}^{2}},\omega_{1}=\sqrt{\frac{R+r_{o}}{{RL}_{o}C_{P}}}\;\\ t>0 \end{array}\right.\;$$

In the above equation, the roots of the characteristic equation were simplified and expressed in the form λ_1,2_ = α_1_ ± jω_d1_, where ω_d1_ represents the damped oscillation frequency of the circuit. The constants of the general solution, A_1_ and B_1_, can be determined based on the initial conditions, while the particular solution, u_C_(t)_P_, can be calculated through the input signal.

At the initial moment, when the driving logic is inactive (t = 0), the parallel capacitor C_P_ and the sensing coil L_o_ are fully discharged. The system thus starts from a zero-state condition, and the rising edge of the step input voltage generates a zero-state response. When the transistor switches on (t = 0_+_), a step voltage U_D_ is applied to the equivalent circuit, where the inductance L_o_ is considered a short circuit and the capacitor C_P_ is treated as an open circuit. Under these conditions, the output voltage u_C_ is influenced only by the series resistance of the inductance r_o_, the current-limiting resistance R, and the applied step voltage U_D_. The particular solution u_C_(t)_P_ of the corresponding differential equation is given by the following:(9)$${u_{C}\;(t)}_{P}\;=\frac{r_{o}}{R+r_{o}}U_{D}$$

In the zero-state condition of the equivalent circuit, the current through inductance is zero and the capacitor is discharged, and the zero-state constraints of the system are as follows:(10)$$\left\{\begin{array}{c} u_{C}\left(t=0_{+}\right)=0\\ i_{L}\left(t=0_{+}\right)=0 \end{array}\right.$$

By substituting the initial condition into u_C_(t) in Equation (8), the constant A_1_ is calculated as follows:(11)$$u_{C}\left(0_{+}\right)={0=A}_{1}+\frac{r_{o}}{R+r_{o}}U_{D}$$

If, in Equation (2a), the initial conditions are substituted, the equation can be rewritten as follows:(12)$${\left.\frac{du_{C}}{dt}\right|}_{t=0_{+}}=\frac{U_{D}-u_{C}\left(0_{+}\right)-Ri_{L}(0_{+})}{RC_{P}}=\frac{U_{D}}{RC_{P}}$$

By substituting the initial conditions into the derivative of the solution, as given in Equation (8), the following expression is obtained:(13)$${\left.\frac{du_{C}}{dt}\right|}_{t=0_{+}}=\propto_{1}A_{1}+B_{1}\omega_{d1}=\frac{U_{D}}{RC_{P}}$$

By combining Equations (12) and (13) for coefficients A_1_ and B_1_, the following constraint is obtained:(14)$$\propto_{1}A_{1}+B_{1}\omega_{d1}=\frac{U_{D}}{RC_{P}}$$

By combining Equations (11) and (14), the constants of the general solution and the solution of the initial system are as follows:(15)$$\left\{\;\begin{array}{c} u_{C}\left(t\right)=\frac{r_{o}}{R+r_{o}}U_{D}+e^{\propto_{1}t}\left(A_{1}\cos\left(\omega_{d1}t\right)+B_{1}\sin\left(\omega_{d1}t\right)\right)\\ A_{1}=-\frac{r_{o}}{R+r_{o}}U_{D}\\ B_{1}=\left(\frac{U_{D}}{RC_{P}}-\propto_{1}A_{1}\right)\frac{1}{\omega_{d1}}\\ t>0 \end{array}\right.\;$$

Starting from the zero-state initial condition and applying a step voltage through the driving logic, the capacitor starts to charge, and a current starts to flow through the inductance. The mathematical expression for inductance current, given in Equation (16), is derived from Equation (2a) by replacing the capacitor charging current as $ic=C_{P}\frac{du_{C}}{dt}$.(16)$$\left\{\;\begin{array}{c} i_{L}\left(t\right)=\frac{U_{D}-u_{c}\left(t\right)}{R}-{C_{P}\propto_{1}u}_{c}\left(t\right)+C_{P}\;\omega_{d1}e^{\propto_{1}t}\left(A_{1}\sin\left(\omega_{d1}t\right)-B_{1}\cos\left(\omega_{d1}t\right)\right)\\ t>0 \end{array}\right.\;$$

When a step voltage is applied, the voltage response and the current through inductance are underdamped sinusoidal waves, as shown in Figure 4.

Because in our setup the coefficient ∝_1_ is negative, the amplitudes of u_C_(t) and i_L_(t) decrease gradually and become zero at t→∞, when all the energy is consumed as losses by the equivalent series resistance. The inductance L_o_ is considered constant, so the energy transfer issue which enhances the consumed energy was not considered. In compliance with Equation (15), the voltage u_C_(t) is shifted with an offset voltage dependent on the sensing coil’s equivalent series resistance, r_o_. If the resistance that appears between the coil loops is neglected ($r_{o}\ll R$), then the output voltage would be centered on 0 V. The oscillation frequency of the output voltage is $f=\;\frac{\omega_{d1}}{2\pi }$. Based on these considerations, it can be estimated that the voltage across the capacitor is zero and the current through the inductance is maximum, $I_{01}$, at half of the oscillation period, when $t=t_{1}\approx \;\frac{\pi }{\omega_{d1}}$, $u_{C}\left(t_{1}\right)\approx 0$. The value of the peak current through the inductance is defined by Equation (17).(17)$$I_{01}=i_{L}\left(t_{1}\right)\approx \frac{U_{D}}{R}\;\left(1+\;e^{\propto_{1}\frac{\pi }{\omega_{d1}}\;}\right)+\propto_{1}C_{P}\frac{r_{0}U_{D}}{R+r_{0}}{\;e}^{\propto_{1}\frac{\pi }{\omega_{d1}}\;},\;I_{01}>\frac{U_{D}}{R}$$

During the discharging process, the capacitor generates a current that flows into the inductor. The peak current through the inductor, $I_{01}$, exceeds the value limited by the resistance R.

This work presents a measurement system designed for distance detection using the principle of inductive sensing, where electromagnetic interaction occurs between a metallic target and the sensor’s primary transducer. The system incorporates a microcontroller as the central processing unit (CPU), which is responsible both for generating a continuous oscillating output signal and for determining the distance based on this signal. At the moment when $u_{C}\left(t_{1}\right)=0$, corresponding to the zero crossing of the voltage in the $L_{o}C_{P}$ circuit, the external voltage source is disconnected. The equivalent circuit of the inductive proximity sensor with the driving source disconnected is shown in Figure 5.

When the driving source is disconnected, then the first stage ends at $t=t_{1}$, and the second stage begins, t′=0. In the second stage, the current flows through inductance and begins to charge the capacitor. Applying Kirchhoff’s Voltage Law, the next equation is obtained.(18)$$r_{o}i_{L}+u_{L}+u_{C}=0$$

Taking into consideration the current $i_{L}=ic$ and substituting current–voltage relationships, the following equation can be obtained:(19)$$L_{o}\frac{di_{L}}{dt}{+r}_{o}i_{L}+\frac{1}{C_{P}}\int i_{L}dt+u_{C0}=0$$

If we consider the capacitor C_P_ discharged at the beginning of this stage (the driving source $U_{D}\;$ is disconnected at zero crossing of voltage $u_{C}\left(t\right)$), the differential equation of the equivalent circuit is as follows:(20)$$L_{o}C_{P}\frac{d^{2}i_{L}}{{dt}^{2}}\;{+r}_{o}C_{P}\frac{di_{L}}{dt}+i_{L}=0$$

The characteristic equation corresponding to differential Equation (20) has the following roots:(21)$$\lambda_{2\;\;1,2}=\frac{-r_{o}C_{P}\pm \sqrt{{\left(r_{o}C_{P}\right)}^{2}-4L_{o}C_{P}}}{2L_{o}C_{P}}$$

To maintain the oscillating response of the circuit in the second stage, when the driving source $U_{D}$ is disconnected, an underdamped response is needed. The roots of the characteristic equation must be a pair of complex conjugates. The parameters of the equivalent circuit shown in Figure 5 must satisfy the following condition:(22)$${\left(r_{o}C_{P}\right)}^{2}-4L_{o}C_{P}<0$$

The above-mentioned constraint establishes a relationship between the elements of equivalent circuit from Figure 5.(23)$$C_{P}<\frac{4L_{o}}{{r_{o}}^{2}}$$

The parallel capacitor C_P_ is selected based on the minimum inductance value within the measurement range. The solution of differential Equation (20), which describes the system’s behavior having the driving source disconnected, is as follows:(24)$$\left\{\begin{array}{c} i_{L}\left(t^{\prime }\right)=\;e^{\propto_{2}t^{\prime }}\left(A_{2}\cos\left(\omega_{d2}t^{\prime }\right)+B_{2}\sin\left(\omega_{d2}t^{\prime }\right)\right)\\ where\;\propto_{2}=-\frac{r_{o}}{{2L}_{o}},\;\omega_{d2}=\sqrt{\frac{1}{L_{o}C_{P}}-{\left(\frac{r_{o}}{{2L}_{o}}\right)}^{2}}\\ or\;\omega_{d2}=\sqrt{{\omega_{2}}^{2}-{\propto_{2}}^{2}},\omega_{2}=\sqrt{\frac{1}{L_{o}C_{P}}}\\ \;t=t_{1}+t^{\prime };\;t^{\prime }>0; \end{array}\right.$$

In the initial state condition of the second stage, the current through inductance is maximum and the capacitor is discharged, and the zero-state constraints of the system are as follows:(25)$$\left\{\begin{array}{c} u_{C}\left(t\prime =0_{+}\right)=0\\ i_{l}\left(t\prime =0_{+}\right)=I_{01} \end{array}\right.\;$$

By substituting the initial condition into equation of the inductance current, in Equation (24), the constant $A_{2}$ is calculated as follows:(26)$$i_{l}\left(t\prime =0_{+}\right)=I_{01}=A_{2}$$

The derivative of Equation (24) is calculated and brought into the initial conditions as follows:(27)$${\left.\frac{di_{L}}{dt}\right|}_{t\prime =0_{+}}=\propto_{2}A_{2}+\omega_{d2}B_{2}$$

Equation (18) is substituted into the initial condition as follows:(28)$${\left.\frac{di_{L}}{dt}\right|}_{t\prime =0_{+}}=-\frac{r_{0}I_{01}}{L}$$

By combining Equations (26)–(28), the constant B_2_ can be determined, and the solution of the differential equation is the following:(29)$$\left\{\begin{array}{c} i_{L}\left(t^{\prime }\right)=\;e^{\propto_{2}t^{\prime }}\left(A_{2}\cos\left(\omega_{d2}t^{\prime }\right)+B_{2}\sin\left(\omega_{d2}t^{\prime }\right)\right)\\ where\;{A_{2}=\;I}_{01}\\ \;B_{2}=\frac{I_{01}}{\omega_{d2}}\;\left(\frac{r_{0}}{L}+\propto_{2}\right)\\ \;t=t_{1}+t^{\prime };\;t^{\prime }>0; \end{array}\right.$$

The voltage from the measurement point is equal to the voltage across the capacitor. The mathematical equation for u_C_ is obtained if, in Equation (18), the inductance current is substituted with the one in Equation (29).(30)$$\left\{\begin{array}{c} u_{C}\left(t^{\prime }\right)={-e}^{\propto_{2}t^{\prime }}\left({A^{\prime }}_{2}\cos\left(\omega_{d2}t^{\prime }\right)+{B^{\prime }}_{2}\sin\left(\omega_{d2}t^{\prime }\right)\right)\\ {A^{\prime }}_{2}=A_{2}{(r}_{0}+L_{0}\propto_{2})+B_{2}{L_{0}\omega }_{d2},\\ {B^{\prime }}_{2}=B_{2}{(r}_{0}+L_{0}\propto_{2})-A_{2}{L_{0}\omega }_{d2},\\ \;t=t_{1}+t^{\prime };\;t^{\prime }>0; \end{array}\right.$$

The negative sign of the voltage from the measurement point defined by Equation (29) suggests that the signal is appended to the signal u_C_, defined by Equation (15), with a negative alternance of the signal. Starting from the zero state, the voltage across the capacitor decreases to a negative value; the next zero value takes place at $t^{\prime }\approx t_{2}=\;\frac{\pi }{\omega_{d2}}$, $u_{C}\left(t_{2}\right)=0$. When the voltage across the capacitor is zero, the current through the circuit reaches the negative maximum value, I_02_. The peak current is obtained by substituting the time into Equation (29) with $t^{\prime }=\;\frac{\pi }{\omega_{d2}}$.(31)$$I_{02}=i_{L}\left(t_{2}\right)={-e}^{\propto_{2}\frac{\pi }{\omega_{d2}}\;}I_{01}$$

When the driving source is disconnected, the second stage ends at time t_2_. The initial current through the inductance will be represented by the current I_02_ if the driving source is reconnected to the circuit. When the voltage across the capacitor is zero $\left(t^{\prime }\approx t_{2}\right)$, the control voltage U_D_ is connected back into the circuit to preserve the sinusoidal shape of the signals. Furthermore, by replacing the initial current $i_{L}\left(0_{+}\right)$ from Equation (12) with I_02_, the equation of voltage from the measurement point can be found using the results of earlier analyses if we consider the control voltage U_D_ connected in the circuit. Lastly, the following relationships can be used to determine the voltage from the measurement point and the current through the inductance based on the driving logic state.
(32)$$\;\left\{\;\begin{array}{c} u_{C}\left(t\right)=\frac{r_{o}}{R+r_{o}}U_{D}+e^{\propto_{1}t}\left(A_{n1}\cos\left(\omega_{d1}t\right)+B_{n1}\sin\left(\omega_{d1}t\right)\right)\;\;\;\;\;\;\;\;\;\;\;\;\;\;\;\;\;\;\;\;\;\;\;\;\;\;\;\;\;\;\;\;\;\;\;\;\;\\ \begin{array}{c} i_{L}\left(t\right)=\frac{U_{D}-u_{c}\left(t\right)}{R}-{C_{P}\propto_{1}u}_{c}\left(t\right)+C_{P}\;\omega_{d1}e^{\propto_{1}t}\left(A_{n\;1}\sin\left(\omega_{d1}t\right)-B_{n1}\cos\left(\omega_{d1}t\right)\right) \end{array}\\ A_{n1}=A_{1},\;{\;\;B}_{n1}=B_{1}-\frac{I_{n-1\;\;2}}{\omega_{d1}},{{\;\;\;\;\;\;\;I}_{n\;\;1}=i}_{L}\left(T_{2}n+T_{1}\left(n+1\right)\right)\;\\ for\;\;t\in \left(T_{2}n\right.÷\left.\left(T_{2}n+T_{1}\left(n+1\right)\right)\right]\;,{\;U}_{D}-on\;state\;\\ u_{C}\left(t\right)={-e}^{\propto_{2}t}\left({A^{\prime }}_{n\;2}\cos\left(\omega_{d2}t\right)+{B^{\prime }}_{n2}\sin\left(\omega_{d2}t\right)\right)\;\;\;\;\;\;\;\;\;\;\;\;\;\;\;\;\;\;\;\;\;\;\;\;\;\;\;\;\;\;\;\;\;\;\;\;\;\;\;\;\;\;\;\;\;\\ i_{L}\left(t\right)=e^{\propto_{2}t}\left(A_{n\;2}\cos\left(\omega_{d2}t\right)+B_{n\;\;2}\sin\left(\omega_{d2}t\right)\right)\;\;\;\;\;\;\;\;\;\;\;\;\;\;\;\;\;\;\;\;\;\;\;\;\;\;\;\;\;\;\;\;\;\;\;\;\;\;\;\;\;\;\;\;\;\;\;\;\;\;\;\;\\ A_{n\;2}={\;\;I}_{n\;1},B_{n\;2}=\frac{A_{n\;2}}{\omega_{d2}}\;\left(\frac{r_{o}}{L}+\propto_{2}\right)\;\\ {A^{\prime }}_{n\;2}=A_{n\;2}{(r}_{o}+L_{o}\propto_{2})+B_{n\;2}{L_{o}\omega }_{d2},\\ {B^{\prime }}_{n\;\;2}=B_{n\;\;2}{(r}_{o}+L_{o}\propto_{2})-A_{n\;\;2}{L_{o}\omega }_{d2},\;\;I_{n-1\;\;2}=i_{L}\left(T_{1}\left(n+1\right)+T_{2}\left(n+1\right)\right)\\ for\;\;t\in \left(T_{1}\left(n+1\right)\right.÷\left.\left(T_{1}\left(n+1\right)+T_{2}\left(n+1\right)\right)\right]\;,{\;U}_{D}-off\;state\\ T_{1}=\frac{\pi }{\omega_{d1}},\;T_{2}=\frac{\pi }{\omega_{d2}}\\ n=1,\;2,\;3\ldots \end{array}\right.\;\;$$(33)$$i_{L}\left(t\right)=\left\{\;\;\begin{array}{c} \begin{array}{c} i_{L}\left(t\right)=\frac{U_{D}-u_{c}\left(t\right)}{R}-{C_{P}\propto_{1}u}_{c}\left(t\right)+C_{P}\;\omega_{d1}e^{\propto_{1}t}\left(A_{1}\sin\left(\omega_{d1}t\right)-B_{1}\cos\left(\omega_{d1}t\right)\right)\\ t>0 \end{array}\\ A_{n1}=A_{1},\;{\;\;B}_{n1}=B_{1}-\frac{I_{n-1\;\;2}}{\omega_{d1}}\\ B_{1}=\left(\frac{U_{D}}{RC_{P}}-\propto_{1}A_{1}\right)\frac{1}{\omega_{d1}}\\ t>0 \end{array}\right.\;$$

If the circuit’s parameters are kept constant, the circuit can oscillate with a constant amplitude if energy is added periodically. Figure 6 presents the mathematical simulation of the circuit in MATLAB modelled using Equation (32), the forced oscillating regime, the representation of the control voltage U_D_(t), the voltage on the capacitor u_c_(t), and the current through the coil i_L_(t).

The real waveforms of the actual circuit are presented in Figure 7.

The two graphs above show that the amplitude, transient response, and oscillation frequencies are similar, which demonstrates the correct approach of the developed theoretical model. The values for the components are L_o_ = 4.7 mH, r_o_ =16 Ω, C_p_ = 22 nF, and R = 1 k.

The spectral representation of the real oscillator output signal u_c_(t) indicates a low-noise sinusoidal waveform with good frequency stability. The measured signal-to-noise ratio (SNR) exceeded 40 dBc relative to the carrier, which is considered excellent [23], as seen in the spectral plot in Figure 8. Since frequency noise directly limits the smallest detectable variations in the oscillation frequency, a high SNR is desirable, as it influences the resolution of the sensor.

According to Equations (8) and (24), the respective T_ON_ and T_OFF_ periods are unequal. If R ≫ r_o_, then R + r_o_ ≈ R. The oscillation period is T = T_ON_ + T_OFF_ = $T=\frac{1}{2\cdot f_{d1}}+\frac{1}{2\cdot f_{d2}},f_{d}=\frac{1}{T}=\frac{\omega_{d1}\cdot \omega_{d2}}{\pi (\omega_{d1}+\omega_{d2})}$. The oscillation frequency depends nonlinearly on the inductance of the coil L_o_, $f_{d}=\Psi (L_{o})$.

### 2.2. Modelling of the IPS Using Eddy Current Principle

If excited with an alternating signal, the sensor coil creates a magnetic field. The presence of metal objects in the vicinity of the coil influences the magnetic field and implicitly the value of the coil inductance. An induced electromotive force will be created in the conductive material in a variable magnetic field, which will produce a current. The principle illustrated in Figure 9 is based on the local induction currents (eddy currents or Foucault currents [24]) that appear in metal parts when they are subjected to a variable magnetic flux (ωt), being a consequence of Faraday’s law.

According to Lenz’s Law, the primary magnetic field produced by the coil $\vec{B_{P}}$ and the secondary one produced by the induced current (eddy current) in the conductive material $\vec{B_{S}}$ are interdependent and opposed. The total magnetic field $\vec{B_{T}}$ in the coil is the vector sum of the inductions of the magnetic fields produced by each current (ip, is):(34)$$\vec{B_{T}}=\vec{B_{P}}+\vec{B_{S}}$$

If we consider the direction as well, then Equation (34) becomes the following:(35)$$\vec{B_{T}}=\vec{B_{P}}-\vec{B_{S}}$$

Since the two vector quantities are in opposition, the resulting field inside the coil will be weakened, the effect being described below. The induced currents are not surface currents; they penetrate the material under test. The penetration depth of the eddy currents is given by the following relationship:(36)$$\delta \;\left[m\right]=\sqrt{\frac{2\rho }{\omega \mu }}$$
where ρ [Ωm] is the resistivity of the material under test, ω=2πf $\left[\frac{rad}{s}\right]\;$ represents the angular frequency, and $\mu =\mu_{r}\cdot \mu_{0}\left[\frac{H}{m}\right]$ is the magnetic permeability of the target material. The inductive sensors based on eddy currents are used in identifying metallic objects, in determining material defects (inhomogeneities), in determining material thicknesses, etc.

A sensor whose operation is based on the production of eddy currents can be modelled with a transformer whose coupling coefficient depends on the distance between the metallic object and the coil, as seen in Figure 10.

The proposed model is composed of two circuits: the primary circuit is characterized by the impedance, $Z_{o}=R_{o}+j\cdot X_{o}$, where R_o_ is the coil resistance, and the complex term, $X_{o}=\omega \cdot L_{o}$, represents the inductive reactance of the coil [20]. When the object under test is far from the coil, the two circuits do not influence each other; the eddy currents do not exist or have a very small value. If a metal object is brought into the vicinity of the sensor coil, eddy currents appear in the metal object, induced by electromagnetic induction. The magnetic fields produced by the two currents interact, and hence the impedance of the primary circuit changes as follows: $Z_{C}\neq Z_{o}$, $Z_{C}=R_{C}+j\cdot X_{C}$. The magnetic field created by the eddy currents opposes and weakens the primary magnetic field, as seen in Figure 9. To determine to what extent the material (sample under test) affects the primary circuit formed by R_o_ and L_o_, Kirchhoff’s second law can be applied to the two circuit loops in Figure 10.(37)$$\left\{\begin{array}{c} V=R_{o}\cdot I+j\cdot \omega \cdot L_{o}\cdot I-j\cdot \omega \cdot M_{1}\cdot I_{e}\;\;\;\;\;\;\;\;\;\;\;\;\;\;\;\;\;\;\;\;\;\;\;\;\;\;\;\;\;\\ 0=R_{e}\cdot I_{e}+j\cdot \omega \cdot L_{m}\cdot I_{e}+j\cdot \omega \cdot L_{1}\cdot I_{e}-j\cdot \omega \cdot M_{2}\cdot I \end{array}\right.$$
where ω=2πf $\left[\frac{rad}{s}\right]\;$ represents the angular frequency of the excitation signal, R_o_ is the resistance and L_o_ the inductance of the primary coil (without the metallic object present), and R_e_ and L_1_ are the resistance and inductance, respectively, of the circuit loop where eddy currents are flowing. The inductance L_m_ is found in the secondary circuit and produces the leakage flux. It also models the magnetic field caused by eddy currents, and it does not reach the L_o_ value. The mutual inductivity between the two coils is $M_{1}=M_{2}=k\sqrt{L_{o}L_{1}}$. The coupling coefficient, k, depends on the distance between the coil and the device under test; it decreases when distance increases and $k\in \left[0÷1\right).$ To highlight the behavior of the sensor coil in the proximity of the metal objects, the impedance of the primary circuit shown in Figure 10 is analyzed. The aim is to identify the influence of the coefficient k on the impedance. From the current–voltage relationship of the primary circuit $Z_{C}=\frac{V}{I}\;$ and Equation (37), the following expression is obtained for Z_C_:(38)$$Z_{C}=R_{o}+j\cdot \omega \cdot L_{o}+\frac{k^{2}\cdot L_{o}\cdot L_{1}\cdot \omega^{2}}{R_{e}+j\cdot \omega \cdot \left(L_{1}+L_{m}\right)}$$

In the previous equation, if the real and imaginary parts are separated, the influence of the sample on the primary circuit, R_C_ and L_C,_ can be determined.
(39)$$R_{C}=R_{o}+\frac{\omega^{2}\cdot k^{2}\cdot L_{o}\cdot L_{1}\cdot R_{e}}{{R_{e}}^{2}+\omega^{2}{\left(L_{1}+L_{m}\right)}^{2}}=R_{o}+R_{r}\;L_{C}=L_{o}-\frac{\omega^{2}\cdot k^{2}\cdot L_{o}\cdot L_{1}\cdot \left(L_{1}+L_{m}\right)}{{R_{e}}^{2}+\omega^{2}{\left(L_{1}+L_{m}\right)}^{2}}=L_{o}-L_{r}$$

The primary circuit is represented by R_o_ and L_o_, according to Equation (37); in the presence of coupling k, R_C_ >> R_o_, and the equivalent resistance increases. The term R_r_ represents the ohmic component of the impedance (in the secondary circuit, R_e_ has the same meaning) reflected in the primary circuit. R_r_ increases if the coupling coefficient increases. The reactive component L_r_, reflected in the primary circuit, has a minus sign, suggesting the second effect of eddy currents, namely that they produce a magnetic field that opposes the primary field (Lenz’s law). Analyzing the evolution of the impedance |Z_c_|, a decrease in the impedance |Z_c_| is observed with increasing coupling k, as seen in Figure 11. However, a decrease in the impedance |Z_c_| determines an increase in the current absorbed by the sensor coil, I, $I=\frac{V}{Z_{C}}$. Once eddy currents appear in the circuit, the electrical power absorbed from the external excitation source increases.

The sensor model based on monitoring the voltage amplitude at the coil terminals, or on monitoring the current absorbed by the oscillating circuit, is addressed in [8,25,26]. In the case of the IPS implemented in [25], the differential structure with two coils is used, of which one is chosen as a reference. To determine the distance, the amplitude of the differential signal is analyzed. When the coil and the target are close together, losses from eddy currents cause the oscillation amplitude to decrease in the amplitude analysis method, an issue noticed also in our experiments. ADC signal conditioning, however, is not an effective technique, as stressed in [25]. The ADC resolution affects the detection accuracy, and the dynamic range of signal amplitude variation varies with distance. Only 27% of the ADC’s dynamic range is utilized.

Monitoring the oscillation frequency is the foundation of the sensor model presented in this paper, as previously demonstrated also in [27,28]. Equation (39) states that the coil inductance is dependent on the target through the circuit parameters R_e_, L_1_, and L_m_, as well as the position in relation to the target through the coupling coefficient k. Consequently, the oscillation frequency varies with the position of the target relative to the coil through the parameter k, $f_{d}=\Psi \left(L_{0}\right)=\Psi \left(L_{C}\right)\equiv \Gamma \left(k\right).\;$ In addition to the primary measure, distance, the oscillation frequency in the suggested sensor model is also influenced by temperature because of variations in the electrical conductivity of the coil resistance R_o_ and the target’s material. It is challenging to find a mathematical formula that captures the relationship between temperature and oscillation frequency. By simulating the mathematical model that describes the oscillation frequency, however, this dependence can be made clear.(40)$$f_{d}=\frac{\omega_{d1}\left(Lc\left(k,T\right)\right)\cdot \omega_{d2}\left(Lc\left(k,T\right)\right)}{\pi \left(\omega_{d1}\left(Lc\left(k,T\right)\right)+\omega_{d2}\left(Lc\left(k,T\right)\right)\right)}=\Pi \left(k,T\right)$$

In Equation (40), Lc is the equivalent inductance of the coils in the presence of magnetic coupling, k is the coupling coefficient, and T is the ambient temperature. Under stationary conditions (k = constant) the influence of temperature on the oscillation frequency is shown in the next section. The circuit model has the parameters presented in Table 1, the target material being copper.

### 2.3. Embedded Hardware Design and Considerations

In this study, the IPS model is based on embedded technology, the full system being set up on an Analog Discovery Studio board, presented in Figure 12. The parallel LC circuit is brought into oscillator mode using a microcontroller. The oscillation frequency depends on the inductance of the magnetic field generator coil (L_o_). When a metallic object (a target) comes into the inductive proximity sensor’s field of detection, an eddy current builds up in the incoming metallic object. This eddy current reacts to the source of the magnetic field, and hence it acts to reduce the inductive sensor’s own oscillation field [29]. As a result, there is a change in the impedance value of the coil, which acts as a magnetic field generator.

The operating model proposed in this paper is based on the variation in the oscillation frequency with distance, using the eddy current. The proposed topology uses embedded technology; the IPS is implemented with a small number of components. The electrical schematic is shown in Figure 13. Processing the signal provided by the L_o_C_p_ oscillating circuit is performed by a transition detection and focusing circuit with a comparator. The circuit formed by resistors R_3_, R_4_, and C_1_ focuses the signal on Vcc/2, this being the common mode voltage for the transition detection comparator.

To bring the circuit to the maintained oscillator operating mode, the microcontroller algorithm must implement the conditions presented in the previous paragraph. Initially, an energy injection (boost energy) is made in the L_o_C_p_ circuit by switching the transistor Q_1_ to the ON state. Subsequently, changes in the output signal $u_{C}\left(t\right)\;$ are detected with the internal comparator. For the most efficient management process and successive connection/disconnection of the Vcc energy source, comparator interrupts are used. The interrupts are generated by the transitions of the output signal $u_{C}\left(t\right)$. Since we have chosen the comparator inputs to be the offset voltage, the alternating signal shifted, respectively, by the offset voltage, and the zero crossings of the output signal are detected. Thus, the Vcc energy source can be connected/disconnected in the circuit without distorting the output signal $u_{C}\left(t\right).$

The main uncertainty sources in time measurement methods with a microcontroller are quantization and trigger noise if reference frequency is stable [30]. Using crystal oscillators with low temperature drift, the reference frequency uncertainty can be significantly minimized. With a microcontroller from the AVR family—Atmel/Microchip—two frequency measurement methods can be implemented, with similar performances. The first method consists in using a time base of frequency much higher than the unknown signal frequency, f_base_ >> f_x_. This feature is implemented in most 8-bit microcontrollers; the working mode is called Input Capture Pin (ICP) or Input Capture Mode (ICM). The second method uses a time base of frequency much lower than the unknown signal frequency, f_base_ << f_x_. In a fixed interval, the oscillations of the unknown signal are counted and the signal frequency is an integer and a multiple of the signal frequency time base f_x_ = N · f_base_; this method is called the Counter Mode (CM) method. Both methods can generate similar measurement performances from the error’s perspective; the differences are given by the duration of the measurement. For frequency measurement with uncertainty $\varepsilon_{fx}\;$ in both methods, the measurement time is an integer and a multiple of T_base_ = 1/f_base_. The duration of a measurement in both methods can be calculated with the following Equations:
(41)$${∆}_{ICP\;fx}=\frac{M}{f_{x}}\;\left[s\right],where\;M=\frac{{f_{x}}^{2}}{\varepsilon_{fx}\cdot f_{base\;ICP}},f_{base\;ICP}>f_{x};\;\mathrm{for}\;\;\;\mathrm{ICP}\;{∆}_{CM\;fx}=\varepsilon_{fx}\;\left[s\right],\;\;where\;\varepsilon_{fx}=f_{base\;CM};\;\;\;f_{base\;CM}<f_{x};\;\;\;\;\;for\;CM$$
where M is an integer multiple of fixed f_x_ periods required to measure the output signal with uncertainty $\varepsilon_{fx}$. For an uncertainty of 0.1 Hz when the base frequency is 16 MHz and the maximum frequency of the sensor signal is 16,500 Hz, the duration of a measurement is ${∆}_{ICP\;fx}\approx 10\;ms$ and ${∆}_{CM\;fx}=10\;s.$ In terms of measurement duration, the ICP method is more efficient. In practice, the measurement system can provide data almost in real time.

Considering internal operations management, the second method, CM, is more convenient, having approximately 10 s available for handling software operations in the microcontroller. In the case of the ICP method, because some routines and functions cannot be executed in 60 µs (the minimum period of the unknown signal), they are fragmented. Their execution is performed sequentially, synchronous with the alternative output signal, having M = 140 execution intervals available.

## 3. Results and Discussion

The performance of the displacement measuring system depending on frequency variation detection was evaluated in this work. According to the literature, statistical methods based on variability analysis can be used to assess the performance of measurement systems [27,31,32,33]. Two series of frequency measurements were carried out under stationary conditions, with the target placed at distances of 2 mm and 5 mm from the coil, respectively. Each series contained 18 samples, $f_{n},\;\bar{n=1:17}.$ The deviation of each value from the mean frequency is given by $\tilde{f\;}\left[Hz\right],\;{is∆f}_{n}=\;f_{n}-\tilde{f}\;\left[Hz\right]$. The data dispersion is illustrated in Figure 14, showing that the values spread within the interval of Δf_max++_ = ± 0.150 Hz. This spread may be attributed either to inaccuracies of the measurement unit or to noise in the oscillator output signal. To quantify the performance of the frequency measurement unit and the reliability of the proposed IPS implemented with a microcontroller, the standard deviation (SD) was used according to [33]. For the frequency series at 2 mm, the SD was 0.085 Hz, while for the series at 5 mm the SD was 0.095 Hz. In both experiments, the SD was also measured using a 1.5 GHz Agilent 53181A frequency counter, considered as the reference. The values obtained in both experiments are close to the reference SD (≈0.052 Hz). The differences between the calculated SD and the reference SD may be attributed to inaccuracies of the microcontroller-based measurement unit.

In addition to the displacement, another influencing parameter considered is temperature [34]; hence, in the mathematical model, each circuit component listed in Table 1 is modelled linearly with respect to temperature. Simulation results show an almost linear dependence of the oscillation frequency on the temperature. The relative nonlinearity error is approximately 0.04% from the reading value, the sensitivity of the oscillation frequency as a function of temperature is $S_{T}=\frac{∆f}{∆T}\approx -8.35\frac{Hz}{{}^{\circ}C}$, and the average sensitivity obtained from the nonlinear characteristic is $S_{T}=\;-8.1\frac{Hz}{{}^{\circ}C}$. It should be noted that, considering a coupling factor of k = 0.5, the results were obtained with the target close to the coil.

Linear approximation is used to compensate for the temperature dependency, a method widely used in the literature [35], and the results are shown in Figure 15. A linear compensation model was adopted, since the copper target used in our experiments exhibits a quasi-linear dependence of resistivity on temperature within the investigated range [36]. The correction equation is as follows:(42)$$\tilde{f}\left(T=25\;{}^{\circ}C\right)=f\left(T=T_{x}\right)+S_{T}\cdot ∆T\left[Hz\right];\;where\;∆T=T_{x}-25\;{}^{\circ}C\;and\;S_{T}=-8.35\left[\frac{Hz}{{}^{\circ}C}\right]$$

In Equation (42), $\tilde{f}$ refers to the estimation of the oscillation frequency considering the temperature compensation. The value is obtained by shifting the frequency from a temperature T_x_ with a factor dependent on the temperature sensitivity and the temperature difference ∆T from the reference temperature of 25 °C: $∆T=T_{x}-T_{ref};T_{ref}=25\;{}^{\circ}C$.

According to Equation (40), there is a dependence on both temperature and distance. In the theoretical analysis, the distance is modelled with the parameter k—the coupling coefficient. The evolution of the frequency with the two parameters is shown in Figure 16a. The linear dependence on temperature and the nonlinear dependence on the coupling are observed. 

From the perspective of the distance sensor, it is important to note that the frequency depends on the distance. Since the coupling cannot be equaled with the distance, real data from the practical experiment are used to observe the behavior of the sensor. Figure 16b shows experimental data that illustrate the dependence between the oscillation frequency and distance and temperature.

The frequency variation range of the output signal $u_{C}\left(t\right)\;$ for the experimental model at an additional dynamic range of displacement of 14 mm is roughly 1200 Hz, which is the same range as for the theoretical model. Since the characteristic is nonlinear, the local sensitivity decreases with distance: at the beginning of the range, $S_{d}=400\frac{Hz}{mm}$, and at d = 14 mm $S_{d}=4\frac{Hz}{mm}$, as seen in Figure 17. The temperature sensitivity for the experimental data is $S_{T}=-8.2\frac{Hz}{{}^{\circ}C}$.

Considering the system stability expressed by the standard deviation, the minimum detectable variation in the distance between the target and the sensor coil can be defined, according to [27], as ${{Distance\;}_{minim\;}=\;SD/S}_{d}$. Near the 5 mm position, where the sensitivity with distance is Sd = 100 Hz/mm and the standard deviation is approximately 0.094 Hz, the sensor exhibits a resolution of about 0.95 µm. At the 10 mm position, the sensitivity is approximately $S_{d}=35\frac{Hz}{mm}$, the resolution being 2.5 µm. In the short-distance range, at 2 mm from the target, the resolution is approximately 0.35 µm.

Within the operating range of the sensor, between 0 mm and 10 mm, the detection bandwidth is defined by the sensor’s operating range of Δf ≈ 1.8 kHz. The sensing element of the IPS and electronic circuits used for the measurement shown in Figure 13 has a frequency noise $S_{f}=\frac{SD}{\sqrt{∆f}}\approx \frac{2.2\;mHz}{\sqrt{Hz}}.$ Under these conditions, around the operating frequency of 15.5 kHz, the minimum detectable variation in the distance between the coil and the target is approximately $\frac{60\;nm}{\sqrt{Hz}}.$

Compared with the performance of similar systems reported in the literature, such as Sensor B presented in [27,37], the achieved resolutions are comparable. The performance is also similar to that of commercially available IPSs. The main advantage of the proposed system, however, lies in its measurement range, which extends to several tens of millimetres. Due to the nonlinear characteristics of the IPS, the resolution exhibits a nonlinear distribution, an aspect also observed for the voltage-response-based IPS according to [17]. Making a comparative analysis with the parasitic sensitivity for the experimental data at 5 mm, the useful sensitivity is much higher than the parasitic sensitivity (at d = 5 mm, $S_{d}\approx 25{\cdot S}_{T}$).

In contrast, at the farthest position from the coil, the parasitic sensitivity is two times greater than the useful sensitivity (at d = 14 mm, $S_{T}\approx 2{\cdot S}_{d}$). At the maximum distance from the sensor, d = 14 mm, a temperature variation of 1 °C produces a frequency variation of approximately 8 Hz, which corresponds to a displacement of 2 mm, from a distance perspective.

Due to the high parasitic sensitivity, it is imperative to accurately measure the temperature. The uncertainty in determining the temperature generates an uncertainty in determining the distance. If we consider the local sensitivities, the distance determination error is $\varepsilon_{d}\;\left[mm\right]=\frac{S_{T}\cdot \varepsilon_{T}}{S_{d}}$. From a practical point of view, a temperature measurement error of approximately $\varepsilon_{T}\approx 0.1\;{}^{\circ}C$ is feasible. Under these conditions, the distance determination error is presented graphically in Figure 18.

Around the distance of 5 mm, the distance determination error is $\varepsilon_{d}\left[mm\right]=4.8\;µm$, the error increasing proportionally with the distance. Performance can be improved by reducing the temperature measurement error, $\varepsilon_{T}$. To highlight the linear temperature compensation process, three series of experiments were performed, with a stepper motor positioning system, at −13.2 °C, 15.8 °C, and 25 °C, respectively. The results are presented in Figure 19.

Using the linear dependence of Equation (42) with the sensitivity coefficient $S_{T}=-8.35\left[\frac{Hz}{{}^{\circ}C}\right]$, we seek to move the series of values obtained at −13.5 °C and 15.8 °C, respectively, towards the 25 °C series through the compensation procedure, as seen in Figure 20. The maximum relative compensation error is roughly 0.12% in relation to the set of values acquired at 25 °C. Even though the temperature compensation error is relatively large at small distances, it is only about 20 Hz as an absolute value. From a distance measurement error standpoint, an uncertainty of 20 Hz results in an error of roughly 0.50 mm, since the sensitivity with distance is $S_{d}=\frac{400\;Hz}{mm}$. If the system is used at distances greater than 2 mm, the compensation error can be greatly decreased. The system is used, for instance, to measure displacements in relation to a point in the 2 ÷ 14 mm range.

Third-order polynomial approximation functions are employed in the prototype model created for calculating distance, converting the output signal’s frequency into distance following temperature compensation using the relation (42). By moving the target in steps of 0.4 mm within the range of 0 ÷ 14 mm, Figure 21 displays the errors obtained with the prototype model. The maximum error is roughly 120 µm overall. In fact, the errors in [38] are of micrometers at small distances (less than 5 mm), but they can reach 300 µm at large distances. When the target is near the coil in our prototype model, the errors are about 52 µm at short ranges. In the range of 1 ÷ 10 mm, the maximum error is 33 µm. As the temperature rises relative to 25 °C, the error increases.

Table 2 summarizes the performance of the prototype in comparison with the other systems, commercial and reported in the literature.

Compared with the systems reported in the literature and with commercial solutions, the prototype system offers a wide measurement range. When relating the measurement range to the resolution, the prototype shows a very good performance. These results can be further improved by increasing the SNR.

## 4. Conclusions

The challenge of our work was to develop an optimized IPS model with low fabrication cost and a high degree of integration. To this end, a simple solution was adopted, combining an LC oscillator with embedded technology based on an 8-bit microcontroller.

Our study intends to demonstrate an effective measurement method for inductive or capacitive sensors by means of an embedded system that senses variations in the oscillator’s resonant frequency. It has been demonstrated that the oscillation frequency depends on both temperature and the coupling coefficient, meaning that frequency decreases with distance, with the local distance sensitivity decreasing significantly as the target moves away from the coil, the sensitivity, *S_d_*, varying from 400 Hz/mm at 0 mm displacement to 4 Hz/mm at 14 mm.

The temperature dependency is quasi-linear, the nonlinearity error being very low (about 0.04%); the temperature sensitivity obtained from the experimental data is approximately −8.2 Hz/°C. Even though the temperature sensitivity is quasi-linear, measuring the temperature with an error of 0.1 °C produces measurement errors of less than 52 µm in the measurement range of 0 ÷ 10 mm, suggesting a very good performance of the system. The performance can be further improved by increasing the SNR of the output signal, and by using higher-order polynomial approximation functions or lookup tables.

To mitigate the temperature sensitivity, a simple linear correction model based on Equation (42) is proposed, which reduces the effect of temperature variations. Applying the compensation method, the maximum relative compensation error is approximately 0.12%, corresponding to only 20 Hz (about 0.05 mm at high sensitivity points). With polynomial approximation and compensation, the prototype model achieves a maximum distance measurement error of ≈50 µm within the 0–10 mm range. In a short range of displacement of 1 ÷ 10 mm, the total error is around 33 µm, comparable or better than the results reported in the literature. At longer distances (>11 mm), the errors rise while staying within reasonable bounds (approximately 120 µm). The prototype system has a strong temperature compensation and can measure displacements with accuracy up to approximately 10 mm, with a resolution of 0.1 um. The proposed sensor model demonstrates the potential for non-contact measurements of micrometric-scale displacements in structural elements, and therefore may be employed for monitoring serviceability-related deformations (e.g., crack opening, differential settlement) in construction applications.

## Figures and Tables

**Figure 1 sensors-25-06258-f001:**
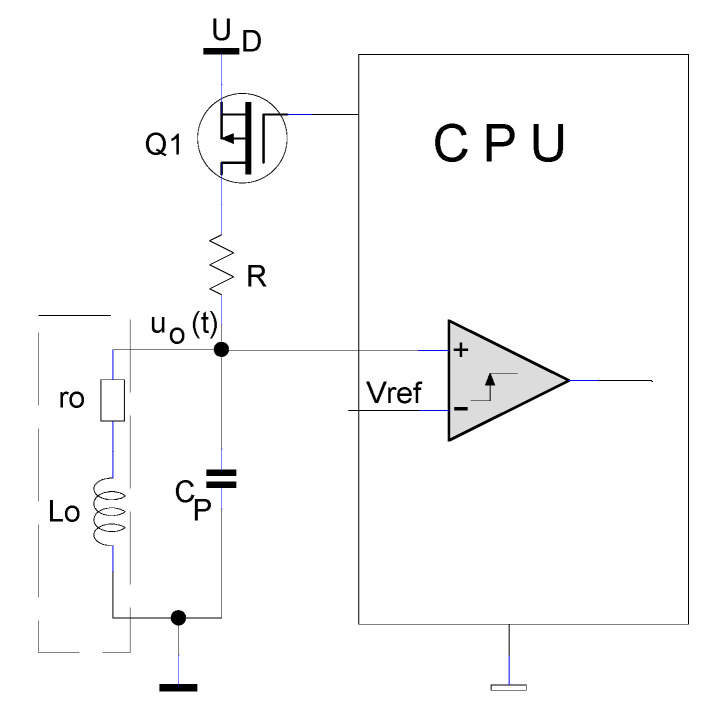
Block diagram of the inductive proximity sensor.

**Figure 2 sensors-25-06258-f002:**
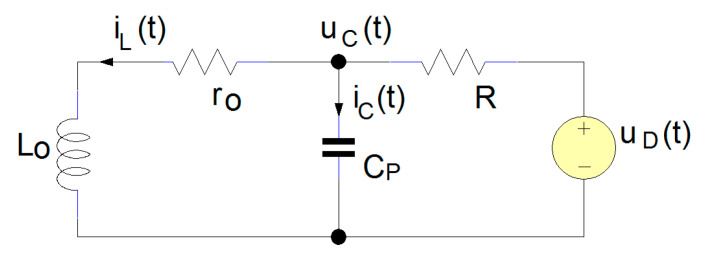
Electrical schematic of the inductive proximity sensor.

**Figure 3 sensors-25-06258-f003:**
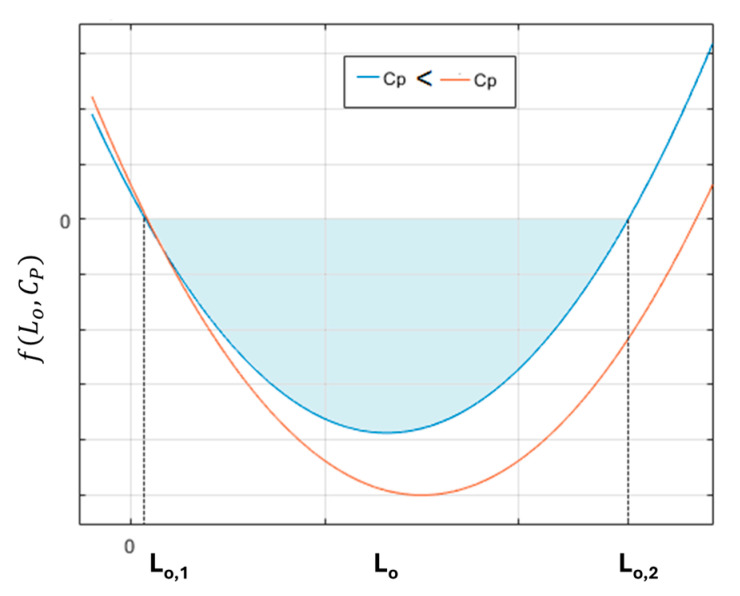
Underdamped condition determined by L_o_ and C_P_.

**Figure 4 sensors-25-06258-f004:**
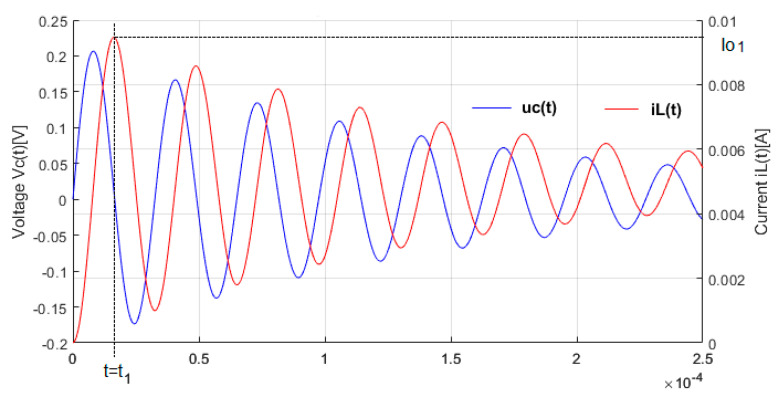
Voltage step response and the current through the inductance.

**Figure 5 sensors-25-06258-f005:**
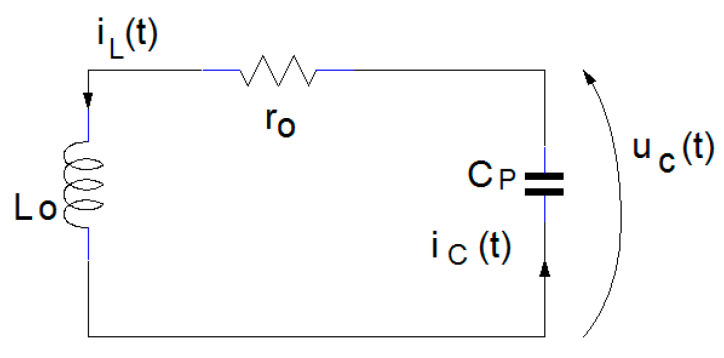
Equivalent circuit of inductive proximity sensor with driving source disconnected.

**Figure 6 sensors-25-06258-f006:**
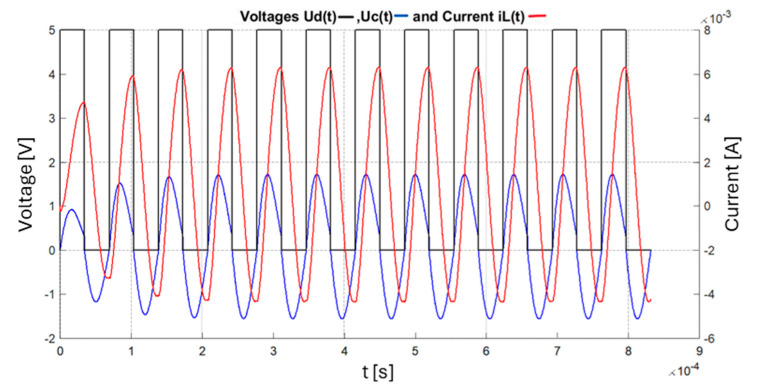
Mathematical modelling of the circuit in MATLAB 2024a.

**Figure 7 sensors-25-06258-f007:**
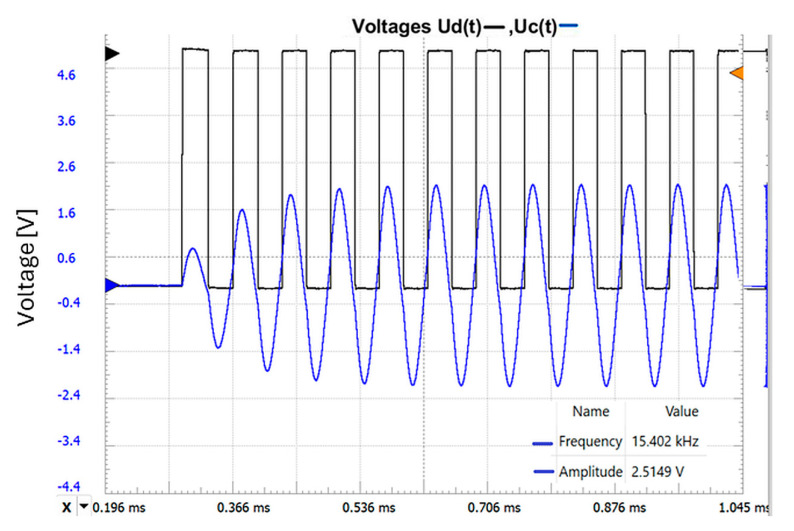
The response of the real circuit, the representation of the control voltage U_D_(t), and the voltage at the terminals of the parallel LC oscillating circuit u_c_(t).

**Figure 8 sensors-25-06258-f008:**
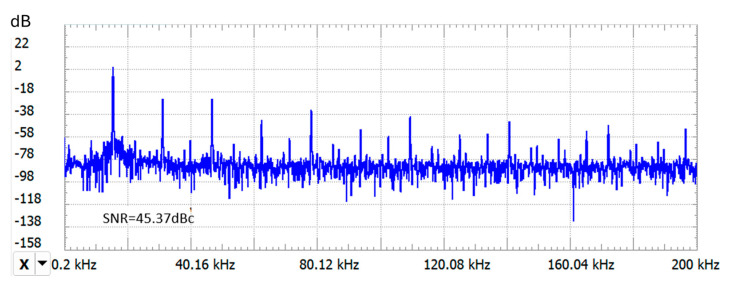
Spectral representation of the u_c_ signal, measured at the oscillator terminals, with the target located 5 mm away from the coil.

**Figure 9 sensors-25-06258-f009:**
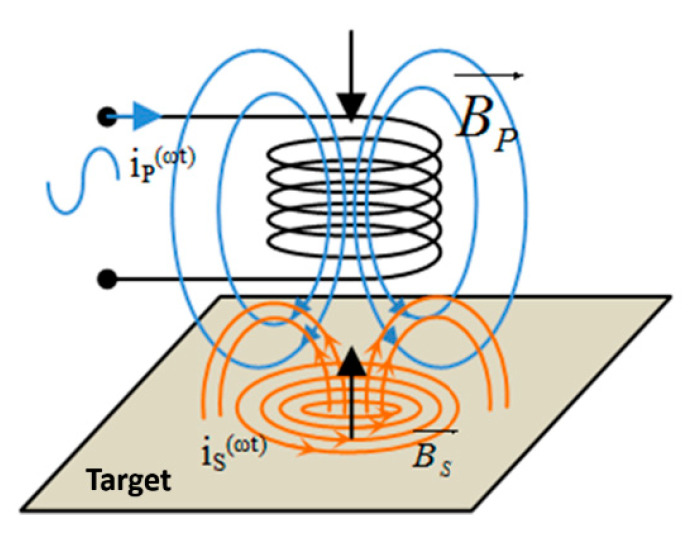
The operating principle of an inductive proximity sensor.

**Figure 10 sensors-25-06258-f010:**
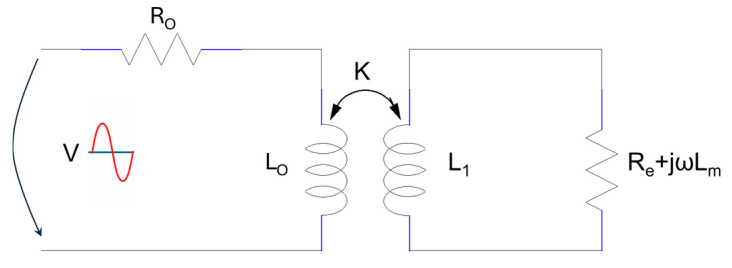
The inductive sensor model, based on eddy currents.

**Figure 11 sensors-25-06258-f011:**
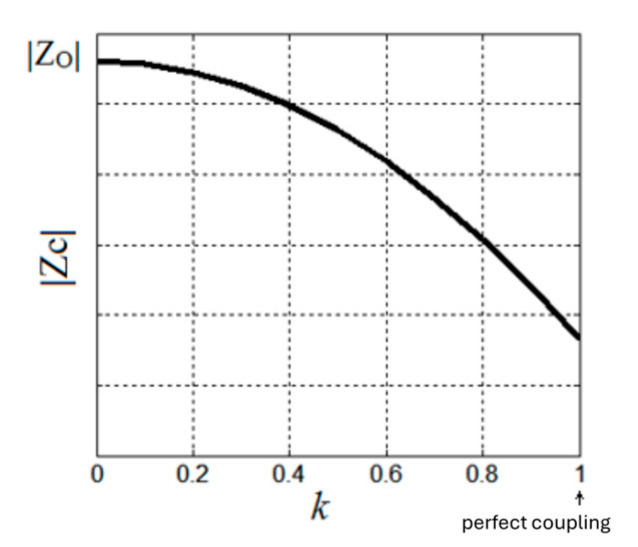
Inductance variation depending on the coupling factor.

**Figure 12 sensors-25-06258-f012:**
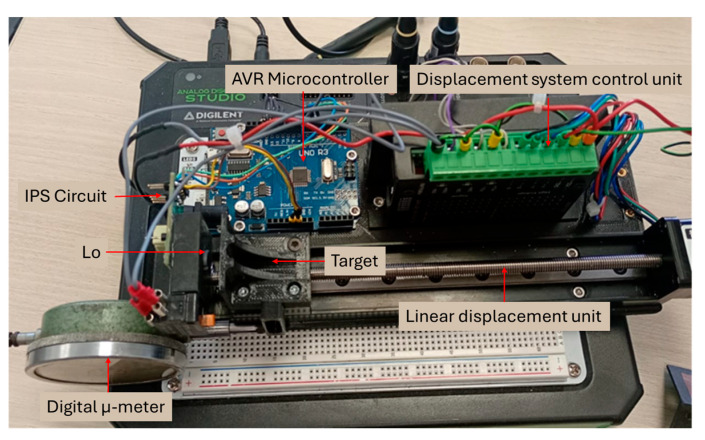
Experimental setup of IPS with displacement system and its control unit.

**Figure 13 sensors-25-06258-f013:**
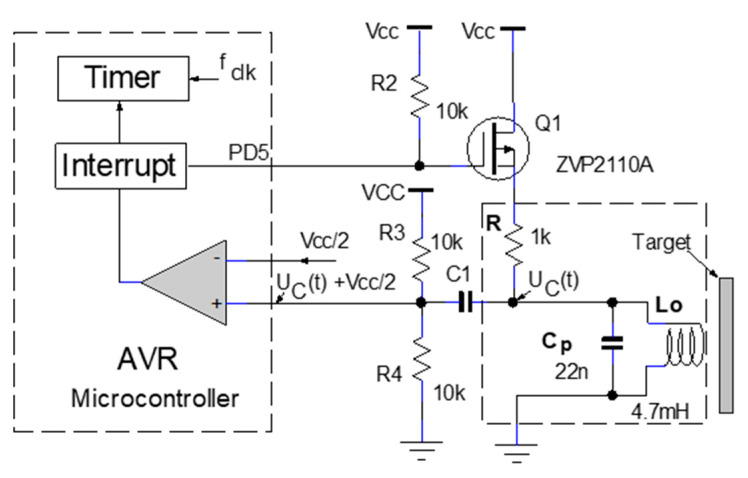
Electrical diagram of IPS implemented with microcontroller.

**Figure 14 sensors-25-06258-f014:**
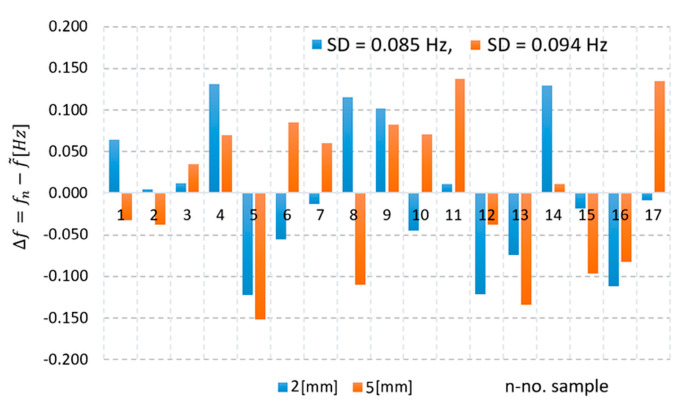
The dispersion of the frequency measurement around the mean values when the target is located at 2 mm and 5 mm from sensing coil.

**Figure 15 sensors-25-06258-f015:**
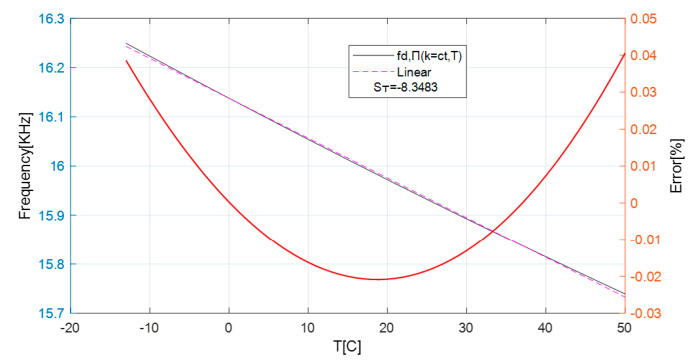
Frequency drift depending on the temperature, linear temperature approximation, and relative error for copper as target material.

**Figure 16 sensors-25-06258-f016:**
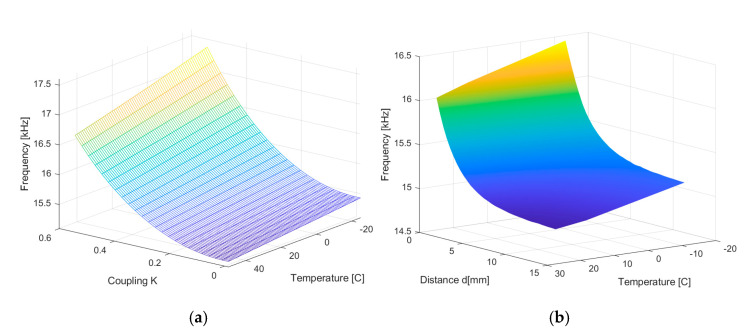
(**a**) Simulated results for the dependence of the oscillation frequency on temperature and coupling coefficient k, and (**b**) experimental results for the dependence of the oscillation frequency on temperature and distance.

**Figure 17 sensors-25-06258-f017:**
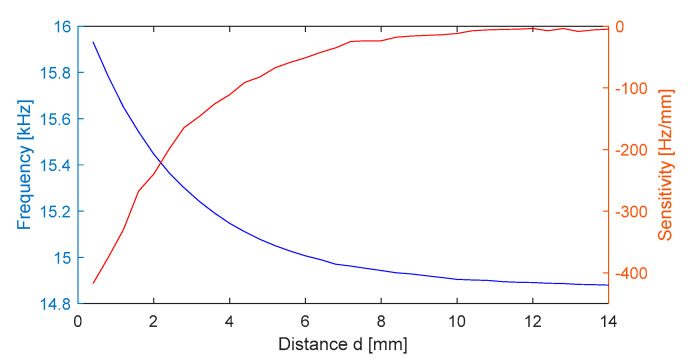
Transducer sensitivity (red) and output signal frequency dependence with distance (blue) for a target of copper at 24 °C.

**Figure 18 sensors-25-06258-f018:**
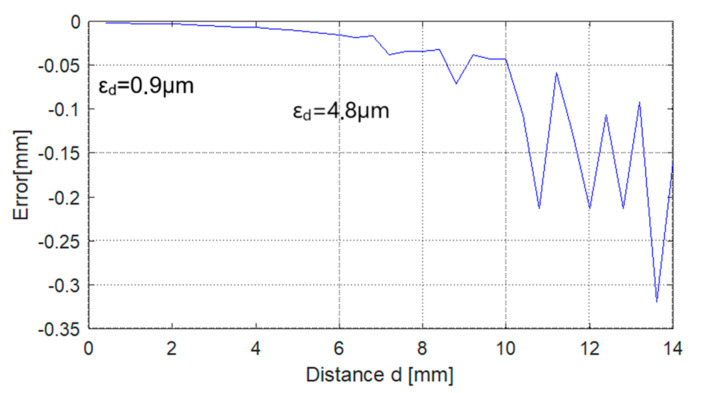
The distance determination error generated by a temperature uncertainty of ε_T_ = 0.1 °C.

**Figure 19 sensors-25-06258-f019:**
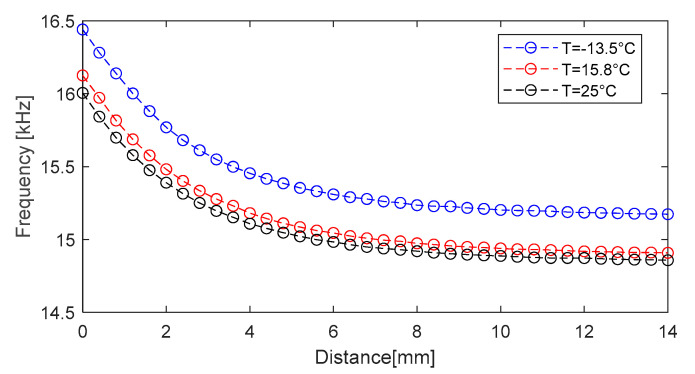
Evolution of the oscillation frequency with distance and temperature.

**Figure 20 sensors-25-06258-f020:**
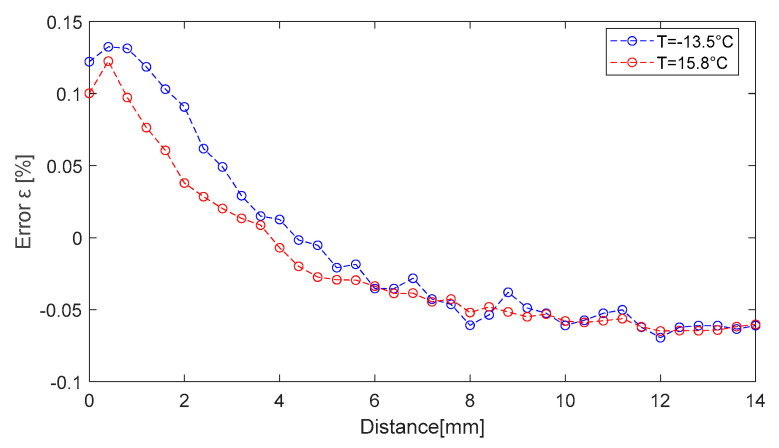
Relative temperature compensation error obtained in two temperature variation experiments.

**Figure 21 sensors-25-06258-f021:**
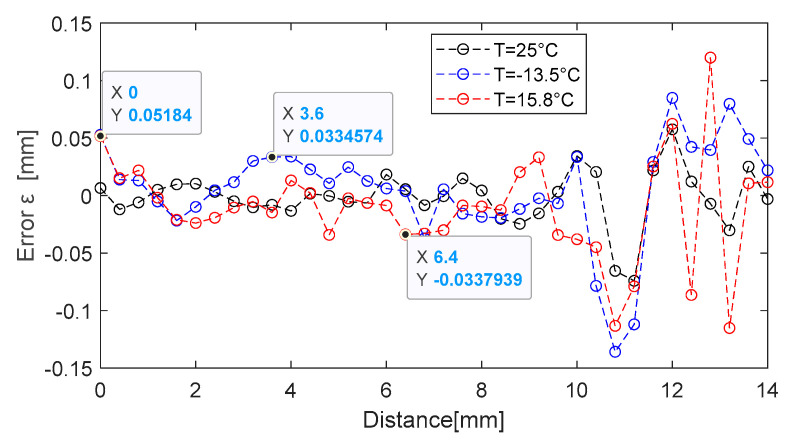
Distance measurement error for the prototype model at ambient temperatures of −13.5 °C, 15.8 °C, and 25 °C.

**Table 1 sensors-25-06258-t001:** Circuit parameters for the electrical circuit model.

Parameter	Value
Coil inductance (L_o_)	4.7 mH
Inductance temperature coefficient (ITC)	350 ppm/°C
Coil series resistance (R_o_)	16 Ω
Coil resistance temperature coefficient (CRTC)	3900 ppm/°C
Parallel capacitor (C_P_)	22 nF
Resistance (R)	1000 Ω
Resistance temperature coefficient (RTC)	−250 ppm/°C
Material eddy current path resistance (R_e_)	0.0025 Ω
Copper target temperature coefficient (CTC)	0.00393/°C
Material eddy current path inductance (L_1_ + L_m_)	0.020 µH
Magnetic coupling coefficient (k)	0.5

**Table 2 sensors-25-06258-t002:** Performance comparison between the proposed prototype and other systems.

Feature	Our Prototype		Baumer IPRM 12I9506/S14	IBS ECA101/ECA110	Sensor B [27]	Sensor Proposed in [37]
Range	5 mm	10 mm	3 mm	1.5–14 mm	4 mm	5 mm
Resolution	0.1 µm	2.5 µm	0.17 µm	8.3 µm	0.634 µm	0.1 µm
Accuracy	±52 µm	±52 µm	±150 µm	25 µm	-	-
Excitation frequency	15 kHz	15 kHz	-	10 kHz–80 kHz	500 MHz	250 Hz

## Data Availability

The data presented in this study are available on request from the corresponding author due to intellectual property (IP) considerations, as they are part of ongoing research and subject to institutional confidentiality and potential patent protection.

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
