# Peer review of "Design and Implementation of an Inductive Proximity Sensor with Embedded Systems"

_sensors, 2025, doi:10.3390/s25196258_

Round 1
Reviewer 1 Report
Comments and Suggestions for Authors
This paper is carefully conceived, thoroughly analyzed, and successfully realized inductive proximity sensor (IPS) of a frequency-output LC oscillator topology. The research is of particular value to the field of embedded sensing and non-contact measurement. The theory analysis is rigorous and fully detailed, experimental confirmation is compelling, and results show an exemplary level of performance (33 µm precision over the 1-10 mm range) competitive with or better than current solutions. The fact that the entire system (excitation circuits, and frequency measurement) is integrated into a single microcontroller chip is the clear strength, revealing a trend towards smaller size, intelligent, and economically viable sensor systems. The discussion of the topic of temperature compensation is of particular interest. In every way, the paper is clearly written and well-structured. It is, in my opinion, a likely choice for publication after some problems identified below are resolved.
My comments are as follows.
- The introduction and state of the art are sufficient yet could be much more fortified to more effectively posit the novelty and contribution of the work here within today's context of cutting-edge sensing technologies. The authors cite mostly old works or seminal works. It would be interesting to see some of the latest (2024-2025) high-impact works to reflect a more thorough involvement with the state-of-the-art and to illuminate the pertinence of your embedded, frequency-based approach. It is strongly recommended to cite the following of the latest and relevant studies
https://www.light-am.com/en/article/doi/10.37188/lam.2023.036; doi: 10.1109/TPEL.2024.3443401; https://doi.org/10.1364/PRJ.525667; doi: https://doi.org/10.1016/j.sna.2023.115003; doi: 10.1109/TCPMT.2025.3533615; doi: https://doi.org/10.1016/j.cej.2025.164787; doi: 10.1109/TIE.2025.3577302; doi: 10.1109/TCPMT.2025.3566026; doi: https://doi.org/10.1002/adfm.202412634.
It is critical that these references be contextually inserted at appropriate points in the manuscript (e.g., introduction and state of the art, results and discussions).
- Some figures (for instance, Figures 7 and 14) exhibit low resolution. It is imperative to provide all figures in high resolution.
- Authors should determine the consistency of subscript display (e.g., L_0 (line 278) and L₀ (in Equation 37)). Consistent use of proper italicization for variables would be useful for clarity.
- There are some spelling errors. Page 17, Equation (42): The comma in -8,35 must be replaced by a dot -8.35 to be compatible with English notation.
- Linear compensation model is observed to perform quite well, as shown. But the authors should say in passing why they can adopt only a simple linear model, especially when others (e.g., that paper that you cite as [25]) adopt non-linear models. Is it the specific operating point selected (i.e., k=0.5) or material properties? Two sentences on that aspect would be helpful.
- The manuscript highlights the correct measurement of a physical quantity, namely distance by analysis of frequency. It would be helpful to augment sections outlining context and future application by including mention of state-of-the-art research in signal processing in fault diagnosis (https://doi.org/10.1016/j.apacoust.2025.110580 and https://doi.org/10.1177/14759217251324671 ) and in non-intrusive monitoring of industry (https://doi.org/10.1016/j.asoc.2024.112445 ). These references will demonstrate the possibility for your sensor to be integrated into comprehensive monitoring regimes and will allow for the advancement of more sophisticated processing algorithms.
Author Response
Comments 1: The introduction and state of the art are sufficient yet could be much more fortified to more effectively posit the novelty and contribution of the work here within today's context of cutting-edge sensing technologies. The authors cite mostly old works or seminal works. It would be interesting to see some of the latest (2024-2025) high-impact works to reflect a more thorough involvement with the state-of-the-art and to illuminate the pertinence of your embedded, frequency-based approach. It is strongly recommended to cite the following of the latest and relevant studies
https://www.light-am.com/en/article/doi/10.37188/lam.2023.036; doi: 10.1109/TPEL.2024.3443401; https://doi.org/10.1364/PRJ.525667; doi: https://doi.org/10.1016/j.sna.2023.115003; doi: 10.1109/TCPMT.2025.3533615; doi: https://doi.org/10.1016/j.cej.2025.164787; doi: 10.1109/TIE.2025.3577302; doi: 10.1109/TCPMT.2025.3566026; doi: https://doi.org/10.1002/adfm.202412634.
It is critical that these references be contextually inserted at appropriate points in the manuscript (e.g., introduction and state of the art, results and discussions).
Response 1: Thank you for your observation, indeed we have cited less than new work, in this respect we have added several citations for studies done in 2024-2025, from the ones that you suggested and some additional ones, as follows:
- references 20-22 (all from 2024) have been inserted in the introduction section and renumbered to match the ascending order
- references 29-37 have been added and inserted where appropriate in the text, 6 of which being from the last 3 years
Comments 2: Some figures (for instance, Figures 7 and 14) exhibit low resolution. It is imperative to provide all figures in high resolution.]
Response 2: [Thank you for the observation, indeed it has been noticed that when zooming on the images, the text was not quite clear. Thus, figures have been exported again from the software where they were captured (e.g. Waveforms and Matlab] with the highest resolution available and have been replaced in the paper.
Comments 3: [Authors should determine the consistency of subscript display (e.g., L_0 (line 278) and L₀ (in Equation 37)). Consistent use of proper italicization for variables would be useful for clarity.]
Response 3: Thank you for pointing it out, you are right, we made some confusions between L_0 and Lo, we have looked through the entire paper and corrected every entry to Lo.
Comments 4: There are some spelling errors. Page 17, Equation (42): The comma in -8,35 must be replaced by a dot -8.35 to be compatible with English notation.
Response 4: Thank you for your comment, indeed there are some spelling errors that have now been corrected.
Comments 5: Linear compensation model is observed to perform quite well, as shown. But the authors should say in passing why they can adopt only a simple linear model, especially when others (e.g., that paper that you cite as [25]) adopt non-linear models. Is it the specific operating point selected (i.e., k=0.5) or material properties? Two sentences on that aspect would be helpful.
Response 5: Thank you for your suggestion, explanation found at line 485-487 was added, basically we adopted a linear model due to material properties: the copper target used in our experiments exhibits a quasi-linear dependence of resistivity on temperature within the investigated range.
Comments 6: The manuscript highlights the correct measurement of a physical quantity, namely distance by analysis of frequency. It would be helpful to augment sections outlining context and future application by including mention of state-of-the-art research in signal processing in fault diagnosis (https://doi.org/10.1016/j.apacoust.2025.110580 and https://doi.org/10.1177/14759217251324671 ) and in non-intrusive monitoring of industry (https://doi.org/10.1016/j.asoc.2024.112445 ). These references will demonstrate the possibility for your sensor to be integrated into comprehensive monitoring regimes and will allow for the advancement of more sophisticated processing algorithms.
Response 6: Thank you, a sentence was added in the conclusions to highlight the applications the authors thought of when developing this study - lines 614-617, and two of the suggested references were added - [22] and [36].
Thank you very much for the appreciations and your support in reviewing our work, your comments have been really helpful!
Reviewer 2 Report
Comments and Suggestions for Authors
The manuscript describe in details the operating principle, the design, and the experimental characterization of an inductive proximity sensor (IPS).
SECTION 1:
In this section (and in the abstract), the authors underline the fact that their IPS is based on the “frequency response” (instead of the “amplitude response” as typically measured). In particular the authors state that “Compared to all the previously mentioned research, our research aims to present an efficient measurement technique for capacitive or inductive sensors, using an embedded system that uses the changes in the oscillator’s resonant frequency for sensing”. The use of the “changes in the oscillator’s resonance frequency for sensing” have been used already in previous IPS sensors and should be cited and eventually compared (both in terms of implementation of the measuring principle as well as in terms of performance). These includes, e.g.,:
Farsi, et al. "500 MHz high resolution proximity sensors with fully integrated digital counter." Measurement 217 (2023): 113045.
V. Matheoud et al. "Microwave inductive proximity sensors with sub-pm/Hz1/2 resolution." Sensors and Actuators A: Physical 295 (2019): 259-265.
More in general, I think the authors should cite and eventually compare the performance obtained with more previously reported IPS sensors, including eventually also some commercially available IPS sensors, particularly those having in their datasheet a clear description of their performance.more previous works on IPS. Previous work on IPS include, e.g.,:
Chaturvedi, et al. "A 19.8-mW eddy-current displacement sensor interface with sub-nanometer resolution." IEEE Journal of Solid-State Circuits 53.8 (2018): 2273-2283.
Zhao, Guangen, et al. "Advances in high-precision displacement and thickness measurement based on eddy current sensors: A review." Measurement 243 (2025): 116410.
Zhao, Guo Feng, et al. "Eddy current displacement sensor with ultrahigh resolution obtained through the noise suppression of excitation voltage." Sensors and Actuators A: Physical 299 (2019): 111622.
SECTION 2 :
It is not very clear to me which parts of the SECTION 2.1 and SECTION 2.2 are important and/or original contributions. I would eventually move same parts of these two sections in an appendix or in supplementary materials. But this is not an important point.
As support for the experimental part, I would definitely cite basic theoretical papers such as Dodd, C. V., and W. E. Deeds. "Analytical solutions to eddy‐current probe‐coil problems." Journal of applied physics 39.6 (1968): 2829-2838.
LINE 147: The values of the inductance and of the resistance of the sensing coil are specified. Please add also the geometrical description of the sensing coil (internal diameter, external diameter, number of turns, pitch, wire width, wire thickness, ....).
LINE 301: What is v in the penetration depth equation ?
LINE 360: “Monitoring the oscillation frequency is the foundation of the sensor model presented in this paper” should be replaced by “Monitoring the oscillation frequency is the foundation of the sensor model presented in this paper, as previously demonstrated also in Refs. [see references mentioned above and probably others]”.
LINES 394-401 (and Table 2): I'm not sure if it is really important to discuss the effect of the magnetic susceptibility of the target material in addition to its resistivity (in the manuscript the measurements are performed only on good conductors with low magnetic susceptibility). If the authors prefer to keep this discussion, than I think that the sentence “Ferromagnetic materials possess a strong magnetization…» is not very clear. “Soft” ferromagnetic materials have almost zero magnetization at zero field but a large magnetic susceptibility whereas “hard” ferromagnetic materials have large magnetization at zero field but almost zero susceptibility. Please specify that you are considering ferromagnetic targets which are made of “soft” ferromagnetic materials (the “hard” one do not change the inductance). I would also mention also that for good conductors which are diamagnetic or paramagnetic, the main change in inductance is due to their resistivity and not to their magnetic susceptibility.
SECTION 3:
I have appreciated the discussion on the temperature cross-sensitivity and on the effect of such cross-sensitivity on the “accuracy” of the distance measurements.
I think it would be also interesting to consider the effect of the frequency (phase) noise on the “resolution”. This can be done computing the ratio between the rms noise of the frequency in a given bandwidth (for example 1 mHz to 1 kHz) expressed in Hz_rms and the sensitivity at a certain distance (for example 4 mm) expressed in Hz/m. This will give the resolution in m_rms. It would be also nice to see a plot of the frequency noise spectral density in the frequency range from 1 mHz to 10 kHz or so.
In this section, it would be certainly useful to compare in more detail the experimentally obtained results for the sensitivity (in Hz/mm) with those obtained with the model described in the previous section.
SECTION 5:
Typo: I guess this section is actually SECTION 4.
Author Response
Comments 1:
In this section (and in the abstract), the authors underline the fact that their IPS is based on the “frequency response” (instead of the “amplitude response” as typically measured). In particular the authors state that “Compared to all the previously mentioned research, our research aims to present an efficient measurement technique for capacitive or inductive sensors, using an embedded system that uses the changes in the oscillator’s resonant frequency for sensing”. The use of the “changes in the oscillator’s resonance frequency for sensing” have been used already in previous IPS sensors and should be cited and eventually compared (both in terms of implementation of the measuring principle as well as in terms of performance). These includes, e.g.,:
Farsi, et al. "500 MHz high resolution proximity sensors with fully integrated digital counter." Measurement 217 (2023): 113045.
V. Matheoud et al. "Microwave inductive proximity sensors with sub-pm/Hz1/2 resolution." Sensors and Actuators A: Physical 295 (2019): 259-265.
Response 1:
Thank you so much for your comment indeed we have missed to refer to some very relevant work in the field, and that has been corrected. To better place the work in the literature, references [20-22] and [29-37] were added, some of them which have been suggested by the reviewer (e.g. ref. 34) and some additional ones (20, 35).
Comments 2: More in general, I think the authors should cite and eventually compare the performance obtained with more previously reported IPS sensors, including eventually also some commercially available IPS sensors, particularly those having in their datasheet a clear description of their performance.more previous works on IPS. Previous work on IPS include, e.g.,:
Chaturvedi, et al. "A 19.8-mW eddy-current displacement sensor interface with sub-nanometer resolution." IEEE Journal of Solid-State Circuits 53.8 (2018): 2273-2283.
Zhao, Guangen, et al. "Advances in high-precision displacement and thickness measurement based on eddy current sensors: A review." Measurement 243 (2025): 116410.
Zhao, Guo Feng, et al. "Eddy current displacement sensor with ultrahigh resolution obtained through the noise suppression of excitation voltage." Sensors and Actuators A: Physical 299 (2019): 111622.
Response 2: Thank you very much for your suggestion, you are right, a comparison of the proposed sensor with some commercial ones and from other studies would be welcomed, hence we have added in the text, a performance comparison table between the proposed prototype and other systems, two commercial sensors and two proposed by refs. 34 and 35.
Comments 3: It is not very clear to me which parts of the SECTION 2.1 and SECTION 2.2 are important and/or original contributions. I would eventually move same parts of these two sections in an appendix or in supplementary materials. But this is not an important point.
Response 3: Thank you very much for your input, indeed sections 2.1 and 2.2 contain the mathematical model of the sensor, the reason we would like to keep it is because we managed to precisely model the whole circuitry mathematically and obtain the same results in Matlab (fig. 6) from the mathematical model and in practice (fig. 7) thus proving strongly the connection between simulation and experiment.
Comments 4: As support for the experimental part, I would definitely cite basic theoretical papers such as Dodd, C. V., and W. E. Deeds. "Analytical solutions to eddy‐current probe‐coil problems." Journal of applied physics 39.6 (1968): 2829-2838.
Response 4 : Thank you, you are, of course, right, reference added (ref. 37).
Comments 5: LINE 147: The values of the inductance and of the resistance of the sensing coil are specified. Please add also the geometrical description of the sensing coil (internal diameter, external diameter, number of turns, pitch, wire width, wire thickness, ....).
Response 5: Thank you very much for your observation, indeed, the geometrical features of the coil should be included, and they have been added between lines 153-155.
Comments 6: LINE 301: What is v in the penetration depth equation ?
Response 6: Thank you very much for noticing, it was a typo and now has been removed: please see line 315.
Comments 7: LINE 360: “Monitoring the oscillation frequency is the foundation of the sensor model presented in this paper” should be replaced by “Monitoring the oscillation frequency is the foundation of the sensor model presented in this paper, as previously demonstrated also in Refs. [see references mentioned above and probably others]”.
Response 7: Thank you very much for your input, we have corrected accordingly and cited the work that you have suggested.
Comments 8: LINES 394-401 (and Table 2): I'm not sure if it is really important to discuss the effect of the magnetic susceptibility of the target material in addition to its resistivity (in the manuscript the measurements are performed only on good conductors with low magnetic susceptibility). If the authors prefer to keep this discussion, than I think that the sentence “Ferromagnetic materials possess a strong magnetization…» is not very clear. “Soft” ferromagnetic materials have almost zero magnetization at zero field but a large magnetic susceptibility whereas “hard” ferromagnetic materials have large magnetization at zero field but almost zero susceptibility. Please specify that you are considering ferromagnetic targets which are made of “soft” ferromagnetic materials (the “hard” one do not change the inductance). I would also mention also that for good conductors which are diamagnetic or paramagnetic, the main change in inductance is due to their resistivity and not to their magnetic susceptibility.
Response 8: Thank you very much, you have indeed made a very good point. We have now come to realize that it does not make very much sense to keep the description of the magnetic behavior of the target, and thus we have removed the whole explanation.
Comment 9:
I have appreciated the discussion on the temperature cross-sensitivity and on the effect of such cross-sensitivity on the “accuracy” of the distance measurements. I think it would be also interesting to consider the effect of the frequency (phase) noise on the “resolution”. This can be done computing the ratio between the rms noise of the frequency in a given bandwidth (for example 1 mHz to 1 kHz) expressed in Hz_rms and the sensitivity at a certain distance (for example 4 mm) expressed in Hz/m. This will give the resolution in m_rms. It would be also nice to see a plot of the frequency noise spectral density in the frequency range from 1 mHz to 10 kHz or so. In this section, it would be certainly useful to compare in more detail the experimentally obtained results for the sensitivity (in Hz/mm) with those obtained with the model described in the previous section.
Response 9: Thank you very much for the appreciation and for the comment. Following this suggestion, we have analyzed the frequency stability of the sensing signal. Instead of computing the resolution as the ratio between the rms frequency noise in a given bandwidth and the sensitivity, we have quantified the frequency noise using the standard deviation of the measured frequency fluctuations (please see inserted text and plot between lines 454 and 473). This approach provides a robust estimate of the effective resolution, directly linking the noise characteristics to the distance measurement performance. The revised manuscript now includes this analysis, together with a comparison between the experimental sensitivity values and those predicted by the model, as suggested, between lines 517 and 530.
Thank you very much for your review and support, all your comments have been incredibly useful!
Round 2
Reviewer 1 Report
Comments and Suggestions for Authors
The authors have satisfactorily addressed all my comments. In my opinion, the paper can be accepted in this present form.
Author Response
Comments 1: The authors have satisfactorily addressed all my comments. In my opinion, the paper can be accepted in this present form.
Response 1: Thank you for the appreciation and for the review!
Reviewer 2 Report
Comments and Suggestions for Authors
I think the authors have very significantly improved the manuscript in many aspect.
There are still a few points which I think has to be improved:
Comment 1 and Comment 7: The authors have introduced one reference (Ref. 34) for the "oscillator frequency variation" IPS. I think that an older one should also be cited at LINE 374:
Matheoud et al. "Microwave inductive proximity sensors with sub-pm/Hz1/2 resolution." Sensors and Actuators A: Physical 295 (2019): 259-265.
Comment 9: the approach of giving the standard deviation of the frequency fluctuations is, in principle, correct and can be used to compute the resolution. However, the detection bandwidth (e.g., 1 mHz to 1 kHz) to has to be specified, otherwise the resolution is meaningless (and the comparison with other sensors is also meaningless). This is because the detection bandwidth affects the standard deviation of the frequency fluctuations.
Author Response
Comments 1:
Comment 1 and Comment 7: The authors have introduced one reference (Ref. 34) for the "oscillator frequency variation" IPS. I think that an older one should also be cited at LINE 374:
Matheoud et al. "Microwave inductive proximity sensors with sub-pm/Hz1/2 resolution." Sensors and Actuators A: Physical 295 (2019): 259-265.
Response 1: Thank you for the suggestion, we have added the suggested reference 38 at line 374.
Comments 2: Comment 9: the approach of giving the standard deviation of the frequency fluctuations is, in principle, correct and can be used to compute the resolution. However, the detection bandwidth (e.g., 1 mHz to 1 kHz) to has to be specified, otherwise the resolution is meaningless (and the comparison with other sensors is also meaningless). This is because the detection bandwidth affects the standard deviation of the frequency fluctuations.
Response 2: Thank you again for the insight, to adhere to your guidelines, we have added a paragraph between lines 524-529. To add an extra explanation, we just want to justify that noise of the experimental IPS sensor introduces uncertainty in the measured position. The circuit elements of the IPS, represented by the the sensor coil as well as the circuitry of the embedded measurement and control system, generate additive measurement noise that perturbs the frequency of the output signal. To highlight the frequency fluctuations under steady-state conditions at room temperature, caused by these noise sources, we employed the standard deviation. In our experiment, the standard deviation integrates the circuit noise over the sensor’s operating bandwidth of 1.8 kHz. By using the standard deviation of the resulting measurements, we determined the resolution of the system in nm/√Hz. We hope that providing these computations, we managed to justify the resolution of the sensor in the operating bandwidth.